# Physics-Aligned Decoding (PAD) for Discrete Protein Structure Representations

**Mhd Hussein Murtada, Z. Faidon Brotzakis, Michele Vendruscolo**[*]
Centre for Misfolding Diseases
Yusuf Hamied Department of Chemistry
University of Cambridge
Cambridge, United Kingdom

## Abstract

Discrete representations learned by deep autoencoders are increasingly reused as intermediate state spaces in generative, conditional, and autoregressive models. In this work, we empirically identify an objective-level failure mode in discrete protein structure tokenizers trained with reconstruction-aligned losses: despite low global reconstruction error, learned tokens encode locally unphysical geometry, including covalent distortions and steric clashes. We show that these violations are deterministic and persistent under reuse. We test the hypothesis that this behavior arises from objective misspecification rather than architectural limitations, and introduce Physics-Aligned Decoding (PAD), a minimal intervention that augments reconstruction objectives with differentiable physical priors. Without changing architecture or regenerating the codebook, PAD reshapes token semantics and restores physical validity while preserving reconstruction fidelity. Our results highlight how loss geometry determines representation semantics, and demonstrate the importance of objective alignment when discrete representations are reused beyond static reconstruction.

## 1 Introduction

Deep representation learning has traditionally emphasized reconstruction fidelity as a proxy for representation quality. In vector-quantized autoencoders (VQ-VAEs) (van den Oord et al., 2017; Razavi et al., 2019), this paradigm enables compression of high-dimensional inputs into discrete latent codes that can be reused for downstream modeling. Recently, this approach has been extended to protein structure, where discrete structural tokens serve as interfaces to generative models, conditional editors, and large language models (Gao et al., 2024; Lin et al., 2025; Yuan et al., 2025).

In these settings, discrete tokens are no longer treated as passive compression artifacts but define a latent state space over which downstream models operate. When representations are reused as intermediate states, errors that are benign under static reconstruction can propagate or amplify, making the geometry induced by the training objective a critical determinant of downstream behavior.

Protein structures provide a stringent test case. Valid conformations lie on a highly constrained physical manifold defined by covalent geometry, steric exclusion, and torsional regularities (Engh & Huber, 1991; Ramachandran et al., 1963; Dunbrack, 2002). Yet most protein tokenizers are trained using reconstruction-aligned objectives, including coordinate-level $L_2$ losses or frame-aligned variants such as FAPE (Jumper et al., 2021), that are insensitive to localized physical violations. As a result, the geometry induced by these losses need not be isometric to the physical manifold on which protein structures reside.

In this work, we show that this mismatch leads to a systematic failure mode: discrete tokens encode unphysical micro-geometry despite achieving low global reconstruction error. We interpret this behavior as a loss-induced geometric misalignment between the learned latent space and the constrained physical manifold. We argue that this behavior reflects objective misspecification rather than architectural deficiency, and we test this hypothesis through controlled intervention.

---

[*]Correspondence: mv245@cam.ac.uk

## 2  RECONSTRUCTION-ALIGNED DECODING AND ITS BLIND SPOTS

Reconstruction-aligned objectives minimize additive losses over geometric features such as Cartesian coordinates, distances, or frames (Ingraham et al., 2019; Gao et al., 2024). These losses induce symmetric penalties that treat all deviations as comparable, regardless of physical interpretation.

In physically constrained systems, this symmetry is problematic. A small displacement into empty space and an equally small displacement into another atom's excluded volume incur similar penalties under an $L_2$ loss, even though the latter configuration is physically forbidden (Bondi, 1964; Chen et al., 2010).

Figure 1 illustrates this failure mode across representative tokenizers. Notably, violations are reproducible under a single deterministic encode–decode pass, indicating that they reflect stable optima of the training objective rather than stochastic decoding noise.

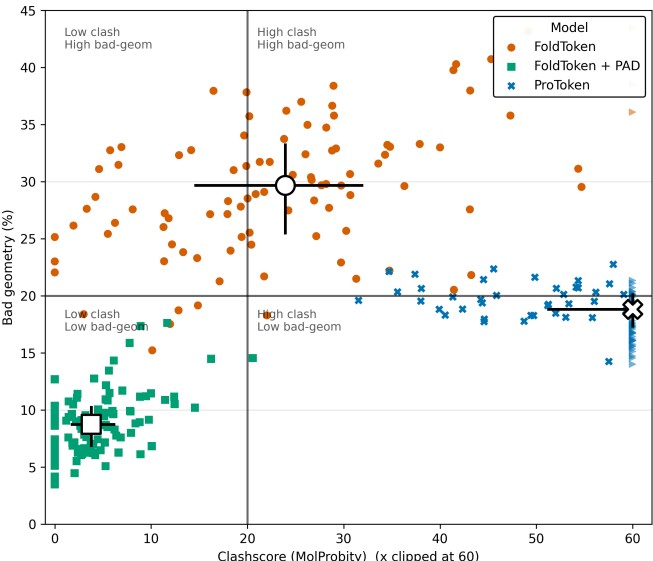

Figure 1: **Deterministic physical violations under reconstruction-aligned decoding.** Clashscore versus bad-geometry fraction for decoded structures on held-out conformations. We define bad-geometry fraction as the proportion of bond and bond-angle instances flagged by MolProbity as outliers (RMSZ > 4) relative to reference covalent geometry. Despite low global error, reconstruction-aligned tokenizers exhibit severe local violations. Physics-Aligned Decoding (PAD) restores physical validity without architectural changes.

## 3  SCIENTIFIC-METHOD FRAMING

We cast this observation as a hypothesis-driven empirical study:

**Observation.** Reconstruction-aligned decoding produces discrete tokens whose decoded structures exhibit local physical violations.

**Hypothesis.** These violations arise from objective misspecification: symmetric reconstruction losses fail to encode asymmetric physical constraints such as steric exclusion.

**Prediction.** If the hypothesis is correct, modifying only the decoding objective, while keeping architecture and codebook fixed, should reshape token semantics and eliminate violations under both single-step decoding and downstream reuse.

**Test.** We implement Physics-Aligned Decoding (PAD) and evaluate physical validity under controlled conditions.

## 4  PHYSICS-ALIGNED DECODING

Physics-Aligned Decoding augments the reconstruction objective with differentiable physical priors that encode local feasibility. The decoding loss takes the form

$$\mathcal{L}_{\text{PAD}} = \mathcal{L}_{\text{rec}} + \lambda_{\text{geom}}\mathcal{L}_{\text{geom}} + \lambda_{\text{vdw}}\mathcal{L}_{\text{vdw}},$$

where $\mathcal{L}_{\text{geom}}$ enforces empirical covalent and torsional constraints (Engh & Huber, 1991; Shapovalov & Dunbrack, 2011) and $\mathcal{L}_{\text{vdw}}$ penalizes steric overlap using smooth van der Waals-inspired terms (Bondi, 1964). Full definitions of these terms are provided in Appendix A.

Crucially, PAD does not alter model architecture or regenerate the discrete codebook. We implement it via parameter-efficient fine-tuning using LoRA (Hu et al., 2021), ensuring that observed changes reflect objective-induced semantic reorganization rather than increased model capacity.

Although the discrete codebook remains fixed, PAD reshapes the effective encoder–decoder mapping rather than acting as an output-space correction. The physical losses are applied to decoded structures, and their gradients propagate through the straight-through estimator to the encoder, shifting token assignment boundaries toward tokens whose decoded geometries satisfy the augmented constraints. PAD therefore reorganizes the distribution of geometries associated with each token without altering the vocabulary itself. Additional training details, LoRA configuration, and data splits are given in Appendix D and Appendix E.

## 5  RESULTS

We evaluate PAD in three complementary regimes: (i) single-step encode–decode reconstruction on held-out conformations, measuring physical validity and reconstruction fidelity; (ii) single-step decoding on out-of-distribution molecular-dynamics (MD) frames, measuring physical consistency on realistic intermediate conformations; and (iii) multi-step autoregressive token reuse, used as a controlled stress test of whether unphysical micro-geometry is encoded directly in token states and accumulates under reuse. This separation distinguishes one-step reconstruction effects from representation-level effects under reuse.

Unless otherwise stated, physical quality is evaluated with MolProbity (Chen et al., 2010). We report clashscore together with bad-geometry fraction, defined as the fraction of bond and bond-angle instances flagged as outliers (RMSZ $> 4$) relative to reference covalent geometry. Because our representation includes backbone atoms and pseudo-$C_\beta$ atoms rather than full side chains, clashscore should be interpreted here as a coarse measure of local steric validity rather than a full all-atom realism metric.

### 5.1  PAD RESTORES PHYSICAL VALIDITY UNDER SINGLE-STEP DECODING

Applying PAD shifts decoded structures into physically admissible regimes (Fig. 1). Reconstruction-aligned decoding exhibits heavy-tailed clashscore distributions and large fractions of covalent outliers despite low global reconstruction error. In contrast, PAD concentrates decoded structures in the low-clash, low-violation regime without changing architecture or regenerating the codebook, supporting the hypothesis that the failure mode arises from objective misspecification.

ProToken is included as a control for whether stronger frame-aligned reconstruction supervision alone suffices to mitigate these failures. Its continued stereochemical violations indicate that improved alignment in reconstruction space is not by itself enough to enforce local physical feasibility.

### 5.2  RECONSTRUCTION FIDELITY IS PRESERVED

To verify that improved physical validity does not come at the expense of reconstruction accuracy, we evaluate both global reconstruction fidelity and local covalent quality on the same held-out conformations. Figure 2 reports bond RMSD versus angle RMSD for decoded structures, with point color indicating $C\alpha$ RMSD relative to the input structure.

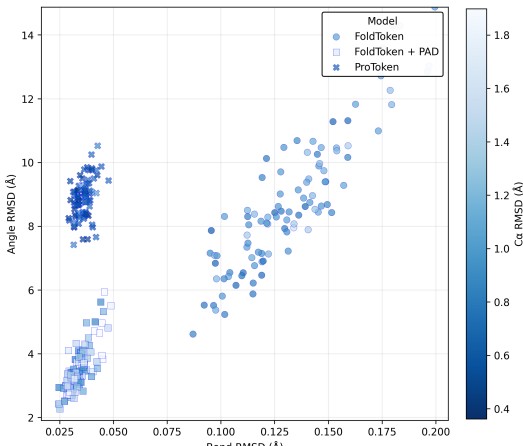

Figure 2: **Reconstruction fidelity is preserved under PAD.** Bond RMSD versus angle RMSD for decoded structures on held-out conformations, with point color indicating C$\alpha$ RMSD to the input structure. PAD moves decoded structures into a regime of substantially reduced bond and angle distortion while maintaining comparable global reconstruction fidelity.

PAD concentrates decoded structures in a regime of substantially reduced bond and angle distortion while maintaining comparable C$\alpha$ RMSD to the reconstruction-aligned baseline. Thus, improvements in physical validity do not arise from sacrificing global reconstruction fidelity, but from reshaping the local geometry associated with discrete tokens.

## 5.3 PAD RESHAPES TOKEN SEMANTICS

To test whether PAD merely smooths decoded coordinates or instead reorganizes the discrete representation itself, we compare token assignments before and after PAD fine-tuning while keeping the codebook fixed. Figure 3 shows the per-frame agreement between baseline FoldToken assignments and FoldToken+PAD assignments.

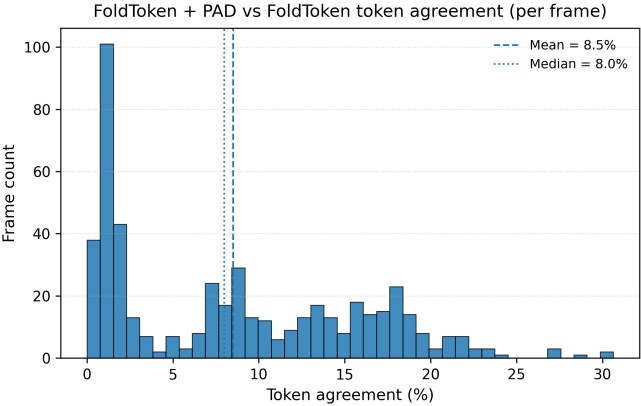

Figure 3: **PAD changes token assignments despite a fixed codebook.** Per-frame token agreement between FoldToken and FoldToken+PAD, defined as the fraction of residues whose discrete token assignment is unchanged. Low agreement indicates that PAD reorganizes token usage rather than acting as an output-space smoother. Additional token-level analyses are provided in Appendix C.

PAD induces substantial token reassignment, indicating that the augmented objective changes the effective latent partition rather than applying a local decoder correction. Because the physical losses act only through decoded structures, this reassignment reflects a shift in encoder decision boundaries via the straight-through estimator and a change in the token–geometry mapping. The improvement

in physical validity therefore corresponds to a semantic reorganization of the latent state space. Structured transition patterns and position-dependent flip rates are reported in Appendix C.

### 5.4 TOKEN REUSE REVEALS SEMANTIC STABILITY

To test whether the improved one-step geometry reflects genuine representational change rather than post hoc smoothing, we evaluate token reuse in a downstream autoregressive setting. We train a lightweight Transformer encoder that predicts next-frame tokens from current-frame token sequences extracted from MD trajectories; the model and training protocol are described in Appendix B.

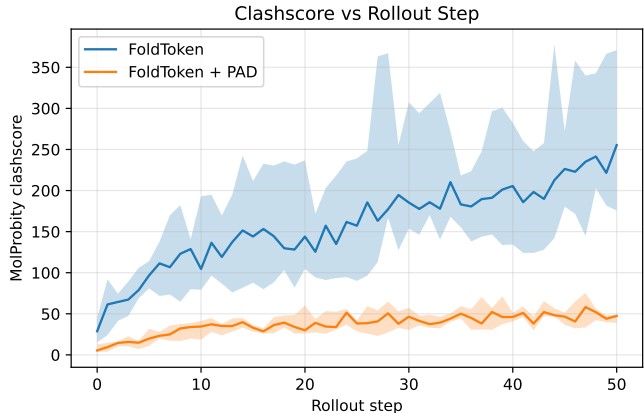

Figure 4: **Token reuse reveals semantic stability.** Median clashscore over multi-step autoregressive rollouts. Reconstruction-aligned tokens rapidly accumulate physical violations, whereas PAD tokens remain stable under reuse.

Figure 4 shows that under autoregressive reuse, reconstruction-aligned tokens accumulate steric clashes rapidly, whereas PAD tokens maintain bounded physical quality. Importantly, improvements are visible at early rollout steps, indicating that unphysical micro-geometry is encoded directly into baseline token semantics rather than arising only from long-horizon accumulation.

The downstream autoregressive model is not intended as a complete generative benchmark or physically faithful dynamics model, but as a controlled stress test of token semantics under reuse. Because architecture, training data, and rollout protocol are matched across conditions, differences in physical validity reflect differences in the semantic content of the tokens themselves.

## 6 DISCUSSION

Our results demonstrate that reconstruction fidelity alone is an insufficient training signal when discrete representations are reused as state spaces. The geometry of the loss function determines which regions of representation space are accessible, and symmetric reconstruction objectives may admit semantically invalid optima in constrained domains.

Unlike post hoc relaxation or inference-time correction, Physics-Aligned Decoding modifies the latent token–geometry mapping itself before downstream learning rather than repairing coordinates after token-space operations have occurred. The resulting improvement in physical validity therefore reflects a change in token semantics rather than an output-space smoothing effect.

While our experiments focus on protein structure, the underlying principle is general: discrete representations learned on constrained manifolds and reused for planning, rollouts, or conditional editing may require objectives that encode feasibility constraints rather than average reconstruction error. PAD enforces local stereochemical feasibility but does not model full molecular energetics or long-range interactions, and evaluation here focuses on backbone plus pseudo-$C_\beta$ representations. The results therefore demonstrate improved physical validity and token reuse stability, while leaving broader characterization of generative diversity and full-atom realism to future work.

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

# A    DETAILED VAN DER WAALS AND STERIC POTENTIALS

This appendix provides the explicit functional forms of the differentiable steric-exclusion and van der Waals terms used in Physics-Aligned Decoding (PAD). The forms are inspired by Rosetta-style Lennard–Jones potentials, but are modified to ensure smooth, numerically stable gradients for end-to-end training (soft-floored distances, a linear continuation at short range, and a smooth cutoff for attraction).

## A.1    SOFT-FLOORED INTERATOMIC DISTANCES

For atoms indexed by $p, q$ with decoded coordinates $\hat{x}_p, \hat{x}_q$, define

$$d_{pq} = \|\hat{x}_p - \hat{x}_q\|.$$

To avoid unbounded gradients as $d_{pq} \to 0$, we use a soft-floored distance

$$\tilde{d}_{pq} = \sqrt{d_{pq}^2 + d_0^2}, \tag{1}$$

with small constant $d_0 > 0$. All Lennard–Jones terms below are evaluated using $\tilde{d}_{pq}$.

## A.2    DIFFERENTIABLE STERIC CLASH PENALTY

A coarse differentiable clash barrier is applied over backbone atoms and pseudo-$C_\beta$ atoms with radii $r_p$. For atom pairs $p < q$, we penalize overlaps using

$$\mathcal{L}_{\text{clash}}(\hat{X}) = \mathbb{E}_{p<q}\Big[\big(\max\{0,\ (r_p + r_q + \delta) - \tilde{d}_{pq}\}\big)^2\Big], \tag{2}$$

where $\delta > 0$ is a small margin. Using $\tilde{d}_{pq}$ (rather than $d_{pq}$) ensures bounded gradients even under severe overlap. This definition is used consistently throughout the paper; the squared term applies to distance penetration quadratically.

## A.3    ROSETTA-INSPIRED VAN DER WAALS ENERGY

We include a smooth van der Waals loss $\mathcal{L}_{\text{vdw}}$ comprising a repulsive (fa_rep) and an attractive (fa_atr) component. For atom pairs $p, q$, let $\sigma_{pq}$ and $\epsilon_{pq}$ denote Lennard–Jones radius and well-depth parameters. Define

$$x_{pq} = \frac{\sigma_{pq}}{\tilde{d}_{pq}}.$$

**Shifted Lennard–Jones form.**    We use the shifted potential

$$E_{pq}^{\text{LJ+}}(\tilde{d}) = \epsilon_{pq}\big(x_{pq}^{12} - 2x_{pq}^6 + 1\big), \tag{3}$$

which is zero at $\tilde{d} = \sigma_{pq}$ and increases sharply for $\tilde{d} < \sigma_{pq}$.

**Repulsive term** (fa_rep).    Let $\tilde{d}_t = 0.6\sigma_{pq}$. For $\tilde{d} \leq \tilde{d}_t$, we use a linear continuation

$$E_{pq}^{\text{lin}}(\tilde{d}) = a_{pq}\tilde{d} + b_{pq},$$

where $(a_{pq}, b_{pq})$ are chosen to match both the value and slope of $E_{pq}^{\text{LJ+}}$ at $\tilde{d}_t$. The repulsive energy is

$$E_{pq}^{\text{rep}}(\tilde{d}) = \begin{cases} E_{pq}^{\text{lin}}(\tilde{d}), & \tilde{d} \leq \tilde{d}_t, \\ E_{pq}^{\text{LJ+}}(\tilde{d}), & \tilde{d}_t < \tilde{d} \leq \sigma_{pq}, \\ 0, & \tilde{d} > \sigma_{pq}. \end{cases} \tag{4}$$

**Attractive term** (fa_atr). Define the standard Lennard–Jones form

$$E_{pq}^{\text{LJ}}(\tilde{d}) = \epsilon_{pq}\big(x_{pq}^{12} - 2x_{pq}^6\big).$$

We apply a smooth cutoff between an inner radius $d_{\text{in}}$ and an outer cutoff $d_{\text{out}}$ using a cubic Hermite switch

$$s(t) = 2t^3 - 3t^2 + 1, \qquad t = \frac{\tilde{d} - d_{\text{in}}}{d_{\text{out}} - d_{\text{in}}}. \tag{5}$$

The attractive energy is

$$E_{pq}^{\text{atr}}(\tilde{d}) = \begin{cases} -\epsilon_{pq}, & \tilde{d} \le \sigma_{pq}, \\ E_{pq}^{\text{LJ}}(\tilde{d}), & \sigma_{pq} < \tilde{d} \le d_{\text{in}}, \\ E_{pq}^{\text{LJ}}(\tilde{d})\, s(t), & d_{\text{in}} < \tilde{d} \le d_{\text{out}}, \\ 0, & \tilde{d} > d_{\text{out}}. \end{cases} \tag{6}$$

In our implementation, $(d_{\text{in}}, d_{\text{out}})$ are chosen as fixed multiples of $\sigma_{pq}$, and the switch ensures continuous energy and gradients at the cutoff.

**Total van der Waals loss.** The full van der Waals loss sums over valid atom pairs, excluding bonded and near-bonded interactions via an exclusion mask $\chi_{pq} \in \{0, 1\}$:

$$\mathcal{L}_{\text{vdw}}(\hat{X}) = w_{\text{rep}} \sum_{p<q} \chi_{pq} E_{pq}^{\text{rep}}(\tilde{d}_{pq}) + w_{\text{atr}} \sum_{p<q} \chi_{pq} E_{pq}^{\text{atr}}(\tilde{d}_{pq}). \tag{7}$$

Together, $\mathcal{L}_{\text{clash}}$ and $\mathcal{L}_{\text{vdw}}$ provide complementary shaping: the clash barrier suppresses forbidden overlaps, while the van der Waals term provides a smooth repulsive/attractive landscape that guides decoded structures toward physically plausible local minima under differentiable optimization.

## B   AUTOREGRESSIVE ROLLOUT MODEL

For the downstream multi-step reuse stress test (Sec. 5.4), we train a lightweight Transformer encoder that predicts the next-frame token at each residue given the current-frame token sequence. Concretely, the model embeds per-residue discrete tokens with an embedding dimension of $d = 256$, adds sinusoidal positional encodings over residue index, and applies $L = 4$ TransformerEncoder layers with $H = 8$ attention heads and feedforward width $4d$, followed by a linear classifier to $K$ token logits per residue. The model is trained with cross-entropy on next-frame prediction pairs $(z_t, z_{t+1})$ extracted from tokenized MD trajectories, where $z_t \in \{0, \ldots, K-1\}^L$ is the full-length residue token vector for a frame. We use an 80/20 split over trajectory frames (first $0.8T$ frames for training, remaining $0.2T$ frames for seeding held-out rollouts), AdamW optimization with learning rate $3 \times 10^{-4}$ and weight decay $10^{-2}$, batch size 16, gradient clipping at 1.0, and train for 10 epochs (dropout 0.1; GELU; pre-norm). The vocabulary size $K$ is inferred from the maximum token id observed in the training frames ($K = \max(z_{\le 0.8T}) + 1$) unless provided explicitly. All rollout experiments use identical model architecture, training procedure, and sampling hyperparameters for FoldToken and FoldToken+PAD, so differences in physical validity arise from token semantics rather than downstream model differences.

## C   TOKEN-LEVEL EFFECTS OF PHYSICS-ALIGNED DECODING

This appendix provides additional token-level analyses complementing the main-text agreement histogram in Fig. 3. While the main paper focuses on per-frame agreement, the visualizations below reveal how the observed semantic changes arise from structured transitions between discrete token states and from localized, position-dependent effects along the sequence.

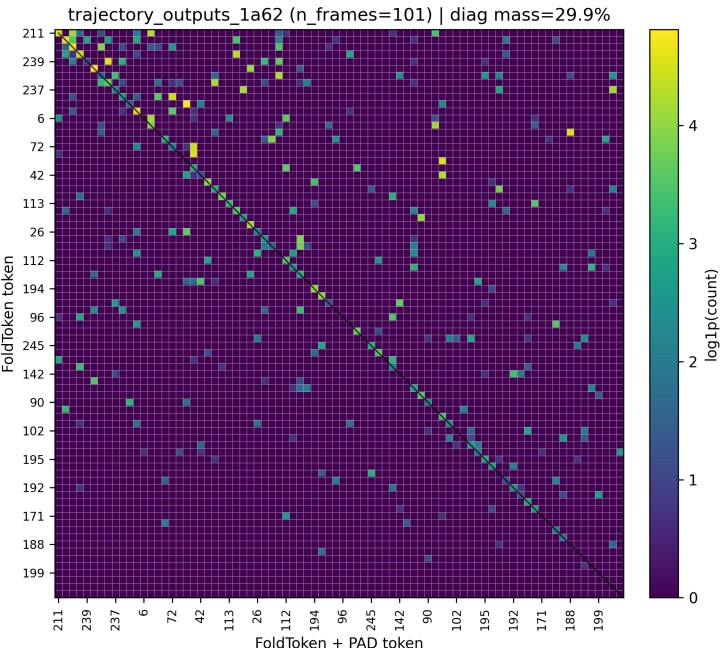

Figure 5: Aggregated token transition matrix between FoldToken and FoldToken+PAD over 101 frames. Each entry counts how often a residue assigned token $i$ under the baseline model is reassigned to token $j$ under PAD. Color indicates $\log(1 + \text{count})$. The transitions are structured rather than diffuse, consistent with controlled semantic reorganization.

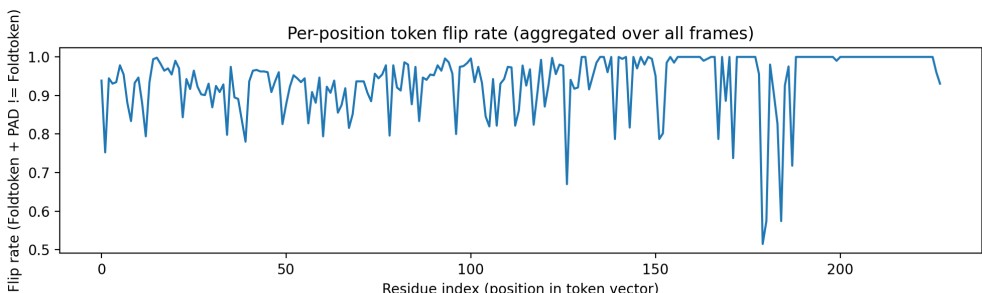

Figure 6: Per-position token flip rate aggregated across all frames, defined as the fraction of frames in which the FoldToken+PAD assignment differs from the baseline FoldToken assignment at a given residue index. While most positions exhibit high stability, several localized regions show consistently elevated flip rates, indicating position-dependent semantic reorganization under PAD.

## D  LoRA FINE-TUNING CONFIGURATION

This appendix summarizes the LoRA configuration used for the geometry-push experiments reported in the main text. We list only the hyperparameters and architectural targets that materially affect the learned representation; standard training, logging, and data-loading details are omitted for brevity.

**LoRA hyperparameters.** We use LoRA with rank $r = 32$ and scaling factor $\alpha = 64$, with dropout 0.1. The base FoldToken model is frozen during fine-tuning, and no bias parameters are trained. LoRA adapters are trained in feature-extraction mode.

**Targeted modules.** LoRA adapters are applied broadly across the encoder and decoder, including attention projections, feed-forward layers, and geometric feature pathways, while preserving the original coordinate prediction heads. Specifically, adapters are attached to:

- the VQ projection layers and embedding MLPs,
- encoder attention ($W_Q, W_K, W_V, W_O$), feed-forward, and geometric feature modules,
- decoder GNN attention, feed-forward, and geometric feature modules,
- decoder coordinate and quaternion prediction projections.

**Optimization and geometry losses.** Fine-tuning uses a learning rate of $1 \times 10^{-4}$ with cosine scheduling and a warmup of 100 steps. Gradients are clipped to a norm of 2.0 and accumulated over 12 steps. Geometry-aware losses are enabled throughout training, including bond length, bond angle, Ramachandran, and $\omega$-torsion penalties (each with weight 0.5), with an additional peptide planarity term (weight 0.05). A van der Waals clash loss is enabled with a small weight (0.01), ramped during training.

## E   TRAINING DATA, SPLITS, AND PROCEDURE

**Training data.** LoRA fine-tuning for Physics-Aligned Decoding is performed on molecular dynamics (MD) trajectory frames derived from the CATH domain dataset (Orengo et al., 1997). Training frames are extracted from equilibrium MD simulations of CATH domains, providing physically realistic intermediate conformations beyond static experimental structures. Only backbone atoms ($N, C_\alpha, C, O$) and pseudo-$C_\beta$ atoms are used during training, consistent with the representation described in the main text.

**Data splits and coverage.** Trajectories are split at the protein level to avoid information leakage across train and evaluation sets. Domains used for PAD fine-tuning are disjoint from those used in all static reconstruction, trajectory-level, and autoregressive rollout evaluations reported in Section 5. For downstream MD-based evaluations, additional out-of-distribution trajectories are drawn from the ATLAS dataset (Vander Meersche et al., 2023), which is never used during fine-tuning.

**Frame sampling.** From each training trajectory, frames are subsampled uniformly in time to reduce temporal correlation and to expose the decoder to a diverse range of physically valid conformations. No frame-level augmentation or artificial perturbation is applied. Each training batch consists of independently sampled frames rather than contiguous trajectory segments.

**Training procedure.** Fine-tuning is performed for a fixed number of epochs over the training frames using the AdamW optimizer. The pretrained FoldToken encoder, decoder, and codebook remain frozen except for the LoRA adapters described above. All reconstruction losses used in the base tokenizer are retained unchanged, with additional physics-aligned terms applied only at the decoding stage. Gradients are propagated through the straight-through estimator to the encoder, allowing token assignment boundaries to adapt without regenerating the discrete codebook.

**Computational cost.** Because PAD is implemented via parameter-efficient fine-tuning, training requires substantially less computation than full retraining. Inference-time decoding incurs only a small constant-factor overhead from evaluating the additional geometry-aware loss terms, with no change to model architecture or codebook size.

