# OpenReview forum: "Physics-Aligned Decoding (PAD) for Discrete Protein Structure Representations"
_ICLR.cc/2026/Workshop/GRaM — ICLR 2026 Workshop GRaM Poster_

### Official Review · Reviewer_Ue2Y · 2026-02-08
**Reconstruction Losses Induce Unphysical Geometry: Physics-Aligned Decoding to Restore Valid Representations**

**Rating:** 8
**Confidence:** 4

**Review:**

## Summary and Problem Statement
This paper highlights a clear and important failure mode of discrete protein structure tokenizers trained with reconstruction-based objectives: despite low overall reconstruction error, the learned tokens deterministically encode locally unphysical geometry, including covalent distortions and steric clashes. As a result, even though the average reconstruction quality looks good, the token semantics are physically incorrect, and these errors accumulate when tokens are reused in autoregressive generative settings.
The authors hypothesize that this problem does not come from architectural limitations, but from objective misspecification. In particular, symmetric reconstruction losses (such as L2 or FAPE) do not capture the strongly asymmetric nature of physical constraints: for example, a small move into empty space and an equally small move into another atom’s excluded volume are penalized in a similar way, even though the second case is physically forbidden.

## Proposed Solution: Physics-Aligned Decoding (PAD)
To test this hypothesis, the authors introduce Physics-Aligned Decoding (PAD), a minimal intervention that keeps the same architecture and the same codebook, but enriches the decoding objective with differentiable physical priors. In practice, the decoding loss becomes: L_PAD = L_rec + lambda_geom * L_geom + lambda_vdw * L_vdw. Here, L_geom enforces geometric constraints (bond lengths, angles, torsions, Ramachandran regions, etc.), and L_vdw penalizes steric clashes using a differentiable term inspired by van der Waals interactions. A key point is that PAD is implemented using LoRA fine-tuning, which means there is no full retraining, no change to the architecture, and no regeneration of the codebook. The paper shows that this simple change at the objective level is enough to reshape token semantics, restore physical validity, and stabilize reuse in generation, while keeping reconstruction quality.

## Scientific Soundness
The main idea is scientifically sound: a symmetric reconstruction loss can indeed produce representations that are globally accurate but locally invalid from a physical point of view. This is a typical case of a mismatch between the metric induced by the loss function and the real constraints of the domain. The paper makes this idea concrete in the context of discrete protein structure tokenization and shows that the geometry induced by the objective directly affects the semantic geometry of the learned representation space.

## Experimental Evidence
The experimental analysis is convincing:
- Physical violations (clashes, poor geometry) appear deterministically after a single encode–decode pass.
- These violations accumulate when tokens are reused in an autoregressive way.
- By changing only the decoding objective (with PAD), these problems can be strongly reduced.
Importantly, the improvements are visible from the very first rollout steps, which supports the idea that unphysical micro-geometry is encoded directly in the token semantics, rather than being introduced by a later decoding artifact.

## Methodology and Evaluation
The methodology is careful and honest.

### Strengths:
1. The architecture is not changed.
2. The codebook is not regenerated.
3. LoRA is used to ensure a minimal and controlled intervention.
4. Comparisons between FoldToken and FoldToken+PAD use:
- the same downstream models,
- the same protocols,
- the same test data.
5. The paper reports:
- single-step decoding effects,
- multi-step rollout behavior,
- token-level analyses (transition matrices, flip rates, etc.).
Overall, this is exactly what one expects from work that aims to show a real change in the semantics of the learned discrete representation, rather than just a cosmetic smoothing or post-processing effect.

## Limitations and Scope
Some limitations are acknowledged and are reasonable given the Tiny Paper / workshop format:
- The study mainly focuses on FoldToken; while this is representative, broader validation on multiple tokenizers would strengthen the conclusions.
- The physical constraints remain local and do not form a full molecular energy model.
- The experimental scope is necessarily limited, and this is not yet a full benchmark study.
These points do not weaken the main contribution, but they indicate that the results should be seen as a focused and well-controlled demonstration, rather than a final universal solution.

## Overall Evaluation

### Pros:
- Clear and well-motivated problem.
- Original and conceptually sound objective-level intervention.
- Minimal and well-controlled experimental setup.
- Convincing evidence that the geometry of the loss shapes token semantics and generative behavior.
- Honest positioning and clear discussion of limitations.

### Cons:
- Evaluation is still limited to a narrow setting (mainly FoldToken).
- Physical modeling is local and approximate.
- Broader validation would be needed for strong general conclusions.

### Verdict
This is a scientifically credible, technically plausible, and experimentally coherent piece of work. While still exploratory and necessarily limited by the Tiny Paper format, it provides a clear and convincing demonstration that the geometry of the loss function shapes the semantics of discrete representations, with concrete and observable effects in generation. The paper is well motivated, methodologically careful, and makes a relevant contribution to the discussion on objective alignment in discrete representation learning for structured physical domains.

**Pmlr Suitability:**

NA

---

### Official Review · Reviewer_UQeW · 2026-02-17
**Review: Objective Alignment Improves Physical Validity, but Evaluation Remains Incomplete**

**Rating:** 3
**Confidence:** 3

**Review:**

## Summary

The paper argues that protein structure tokenizers trained with coordinate reconstruction losses produce locally unphysical geometries (e.g., steric clashes), despite low global reconstruction error. The authors attribute this to objective misalignment: symmetric L2 losses do not encode asymmetric physical constraints such as excluded volume. They propose **Physics-Aligned Decoding (PAD)**, which augments the decoding objective with differentiable physical priors (bond length, torsion, and steric penalties). The modification is implemented through a redefinition of the loss function with LoRA-based fine-tuning, avoiding architectural changes or codebook regeneration. Empirical results show reduced MolProbity clashscore and improved autoregressive rollout stability.

The core idea is interesting and relevant to geometry-aware learning. However, the empirical validation and discussion are currently insufficient to fully support the strongest claims.

---

## Strengths

- **Well-motivated problem.** Highlighting local physical violations in discrete protein structure tokenizers is relevant and potentially impactful. The results suggest that objective design influences geometric fidelity.
- **Clear hypothesis-driven framing.** The paper proposes a causal explanation (objective misspecification) and tests it via a targeted intervention.
- **Minimal modification.** PAD isolates the objective as the main variable (no architectural changes; LoRA-based fine-tuning), strengthening the causal argument.

---

## Weaknesses

### 1. Reconstruction fidelity is not quantitatively demonstrated.

The paper claims that PAD preserves reconstruction accuracy but reports only MolProbity clashscore and “bad-geometry fraction.” These are physical validity metrics rather than reconstruction metrics.
Direct reconstruction measures (e.g., RMSD, lDDT, TM-score, FAPE, distance error distributions) on the same held-out conformations should be reported to substantiate the claim that fidelity is maintained.

### 2. “Bad geometry” is insufficiently defined.

Figure 1 introduces “bad-geometry fraction,” but it is unclear how this metric is computed, what thresholds are used, and how it relates to MolProbity metrics. A precise and reproducible definition is required.

### 3. Limitations of MolProbity in this setup.

The model is trained and evaluated on backbone + pseudo-Cβ atoms (appendix). MolProbity is primarily designed for all-atom validation. Without side chains, steric clashes may be underrepresented or distributed differently. This weakens the interpretation of clashscore as a comprehensive physical realism metric.

### 4. Limited comparison to related approaches.

The paper references some prior tokenizers but omits discussion of widely used structure-based encodings (e.g., FoldSeek) and other geometry-aware or constraint-based strategies.

It remains unclear whether the improvements are novel relative to simpler alternatives such as:
- post-hoc energy minimization,
- inference-time steric penalties,
- geometry-aware training objectives already explored in prior work.

### 5. Dataset selection.

The experimental setup relies on older structural benchmarks. Validation on more recent and diverse protein datasets would strengthen the generality of the conclusions.

### 6. Methods and discussion lack clarity.

Key implementation details are relegated to the appendix, which is not cited in the main text. The paper does not clearly distinguish between:
- single-step encode–decode reconstruction of static structures,
- reconstruction along MD trajectories,
- autoregressive rollout degradation.

Dataset sizes, splits, and evaluation protocols should be explicitly stated.

### 7. Formatting and presentation issues.

- Appendix sections are not cited in the main text.
- Some references appear broken or inconsistently formatted.
- Ablation over individual PAD loss components is limited.
- The claim of token “semantic reshaping” is not rigorously characterized.

---

## Overall Assessment

The idea is promising, however, the current empirical evidence does not fully substantiate the strongest claims, particularly regarding preservation of reconstruction fidelity and novelty relative to existing geometry-aware approaches. The paper would benefit from clearer evaluation protocols, stronger quantitative reconstruction metrics, broader baselines, and more rigorous discussion. Moreover, improving its writing quality, clearness of the methods and amplify the discussions would be necessary.

**Pmlr Suitability:**

NA

---

### Official Review · Reviewer_9XcX · 2026-02-23
**Elegant Physical Correction for Protein Tokenizers, but Downstream Evaluation Falls Short**

**Rating:** 6
**Confidence:** 4

**Review:**

**Summary**

This paper identifies that discrete protein tokenizers trained with reconstruction-aligned losses often encode unphysical micro-geometry (e.g., steric clashes) despite low global errors. To address this, the authors propose Physics-Aligned Decoding (PAD), which incorporates differentiable physical priors into the decoding objective. Using LoRA fine-tuning, PAD restores local physical validity by reshaping token semantics without altering the model architecture or codebook.

**Strengths**

1) Clear Objective: Accurately identifies and addresses a specific, overlooked issue (objective misspecification causing micro-geometry violations) in 3D molecular representation.

2) Intuitive Methodology: Incorporating differentiable physical priors directly into the loss function is a physically grounded and elegant solution.

3) Resource Efficiency: Using LoRA for fine-tuning avoids the high computational cost of retraining the entire autoencoder or regenerating the codebook.

**Weaknesses**

1) Misaligned Downstream Evaluation: The evaluation relies solely on autoregressive (AR) rollouts of molecular dynamics trajectories. It fails to test performance on intended generative tasks. It remains unclear if strict steric constraints compromise the model's generative diversity.

2) Limited Downstream Evaluation Coverage: Evaluation is restricted to MD next-frame AR rollouts and does not test PAD on intended protein design/generation tasks. The impact on generative diversity remains unquantified, and stronger steric constraints may trade off diversity for validity.

3) Typographical Errors: There is an unresolved cross-reference in Appendix B ("For the downstream multi-step reuse stress test (Sec. ??)...").

**Pmlr Suitability:**

NA

---

### Meta-Review · Area_Chair_1Djy · 2026-02-26

**Decision:**

Accept

**Metareview:**

Reviewers were divided, but overall agreed that the proposed idea is sound and addresses a relevant problem in a simple way. Main weaknesses are insufficient evaluation, lack of clarity and presentation issues. These are not critical for an extended abstract, but the authors are still expected to take the feedback into account.

**Relevance To Proceedings:**

Tiny paper — does not apply

**Relevance To Workshop:**

Yes — suitable for GRaM

---

### Decision · Program_Chairs · 2026-03-02

Accept (Poster)